# Aerosol indirect effects in complex orography areas: a numerical study over the Great Alpine Region

Anna Napoli[1,2], Fabien Desbiolles[3,4], Antonio Parodi[1], and Claudia Pasquero[3,5]

[1]CIMA Research Foundation, Savona (Italy)
[2]Università degli studi di Genova, Genova (Italy)
[3]Università degli studi di Milano-Bicocca, Milano (Italy)
[4]Osservatorio Geofisico Sperimentale, Trieste (Italy)
[5]ISAC-CNR, Torino (Italy)

**Correspondence:** anna.napoli@cimafoundation.org

**Abstract.** Aerosols play a crucial role in climate through different feedback mechanisms, affecting radiation, clouds and air column stability. This study focuses on the altitude-dependence of the cloud mediated indirect effects of aerosols in the Great Alpine Region (GAR), an area characterised by high pollution levels from anthropic activities in the Po Valley and a complex orography with some of the highest mountains in Europe. Using a regional atmospheric model, 5-years long convective-permitting sensitivity experiments have been run with different surface aerosol fluxes. The results show that seasonal mean cloud cover, temperature, and precipitations are affected by the aerosol concentrations in the air column, and that the response to pollution is both elevation and season dependent. The overall cloud cover increase with aerosol levels leads to either surface cooling or warming depending on the surface albedo (snow covered or not). Furthermore, different types of clouds have a different response: while the lifetime of low pressure system clouds and orographic clouds is generally increased at high levels of aerosols, convective clouds (typical of the summer season) can decrease at high levels of pollution, due to the reduction of strong updrafts associated with an increased air column stability.

## 1 Introduction

Aerosols affect both directly and indirectly the Earth's radiative budget and climate. As a direct effect, aerosols interact with radiation either through scattering or absorption (Haywood and Shine, 1995). The scattering of solar radiation by aerosol particles typically results in a cooling of the ground surface, while absorption of solar radiation determines local heating of the atmosphere. Moreover, as an indirect effect, in the lower atmosphere aerosols alter the microphysical and radiative properties of clouds acting as cloud condensation nuclei (CCN) and ice nuclei (IN) upon which cloud droplets and ice crystals form. Thus they can influence cloud optical properties, cloud cover, cloud lifetime, and precipitation (Albrecht, 1989; Ramanathan et al., 2001; Rosenfeld et al., 2008). Furthermore, the strong absorption of solar radiation by dark aerosols, such as black carbon, can lead to changes in cloud cover and liquid water content by heating the cloud and the environment within which the cloud forms; this is called the semi-direct effect (Hansen et al., 1997), that may have a significant warming impact on climate by 'burning off' low clouds that scatter solar radiation back to space, but have little impact on outgoing longwave radiation. Thus in the

atmosphere, there is a mixture of scattering and absorbing aerosols, and their net effect on Earth's energy budget is dependent on surface and cloud characteristics.

In the middle of Europe, the Great Alpine Region (GAR) is characterized by its peculiar geographical shape and by complex orography, that are a meteorological trap for atmospheric pollutants emitted in the surrounding lowlands (Schroeder et al., 2014). Depending on orographic and meteorological conditions, pollutants emitted in the densely populated GAR spread in the region and lead to very high concentration of aerosols, with a strong gradient between low and high elevations (Sandrini et al., 2014).

The rate of warming observed over the last decades is elevation-dependent (Pepin et al., 2015), possibly also due to aerosols (Rangwala et al., 2010), which vary with altitude (in concentration and type) and affect the local sensitivity to large scale changes (Stjern et al., 2020). Aerosol effects could also extend to precipitation (Rosenfeld et al., 2008): the radiative effects of aerosols on clouds mostly act to suppress precipitation, because they decrease the amount of solar radiation that reaches the surface, increasing regional atmospheric stability (Zhang et al., 2020). Aerosols also have important microphysical effects

on precipitation (Tao et al., 2012; Fiori et al., 2014): increased Cloud Condensation Nuclei (CCN) slow the conversion of cloud droplets into raindrops (Jonas and Mason, 1974; Rosenfeld, 2000; Borys et al., 2003; Thompson and Eidhammer, 2014). This effect has been shown to generally decrease precipitations (Ochoa et al., 2015), but several and sometime contrasting differences have been described in the literature, mainly depending on precipitation rate and environmental conditions (Qian et al., 2009; Alizadeh-Choobari and Gharaylou, 2017; Alizadeh-Choobari, 2018; Li et al., 2011).

While direct effects of aerosols on the climate system are in general understood and quantified, the quantification of indirect and semi-direct forcing by aerosols is especially complex (Penner et al., 2001). Currently, they are considered one of the most uncertain forcing in climate (Zhang et al., 2016). Although a number of studies on the local effect of pollutants has been published (e.g. Pavlidis et al., 2020), information about the climatological effects on the meteorological conditions of aerosols in complex orography areas at convective-permitting scale is lacking. In order to develop a better understanding of the climatic

role that aerosols play, in this paper we focus our attention to the indirect effects of aerosols only, using the Weather Research and Forecasting Model.

## 2  Methods

In this work we use the Weather Research and Forecasting (WRF) Model (version 3.9.1.1) in non-hydrostatic configuration. Two 5-years long simulations have been run with initial and boundary conditions provided by the Earth System Model EC-

Earth in a historical scenario on a 25 km horizontal grid (Davini et al., 2017).

The study area is the Great Alpine Region, which is represented on two grafted domains shown in Fig. 1: the larger one, with a grid spacing of 12 km, ranging from about 41°N to 51°N latitude and from 0°E to 23°E longitude, and an inner one, with a grid spacing of 4 km, from about 43°N to 49°N latitude and from 3.5°E to 19.5°E longitude. The outer domain includes a convection parameterization, necessary to account for the vertical motions not explicitly represented at the 12 km resolution

that characterize the summer climate in the area under analysis. In the inner domain (4 km), no convective parameterization

scheme is active. The simulations have been run with the two-way nesting approach. The vertical structure of both domains consists of 50 terrain-following levels with a top pressure level set at 50hPa. The vertical resolution is finer near the ground (order of metres), while it is coarser aloft (order of several hundreds of metres). This configuration has already been used in previous works that compared results with lower resolution models and observational datasets (Pieri et al., 2015), indicating that it is an acceptable compromise between the computational burden and the need of resolving smaller scales to predict events with small temporal and spatial scales (Adinolfi et al., 2021; Takayabu et al., 2022).

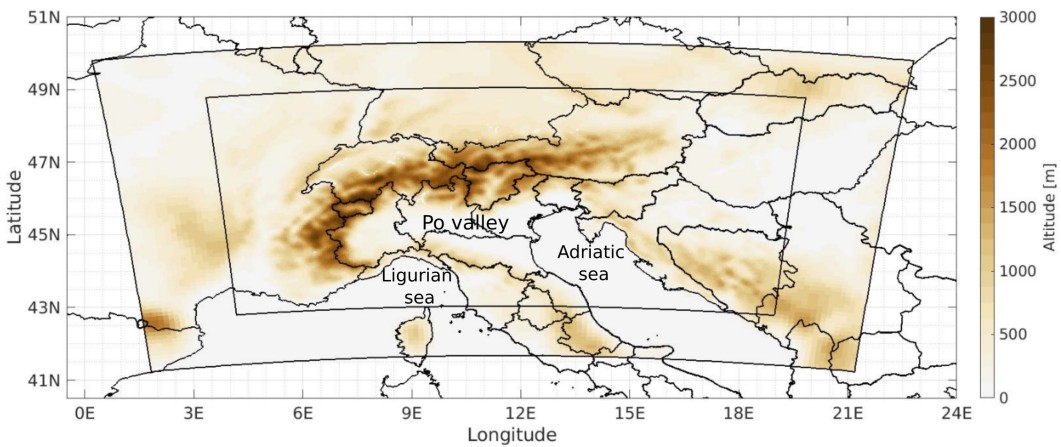

**Figure 1.** *Topography in the two domains used for the 5 years long runs: outer domain at 12 km and inner domain at 4 km grid spacing. Three geographical areas Ligurian Sea, Adriatic Sea and Po Valley are labelled in the figure.*

The Planetary Boundary Layer (PBL) is parametrized with the Yonsei University Scheme (Hong et al., 2006). The Shortwave and Longwave option used is the RRTMG (Iacono et al., 2008). The outer domain includes the Tiedtke Scheme (Tiedtke, 1989) as the convection parametrization. As microphysics scheme the Thompson Aerosol Aware has been used (Thompson and Eidhammer, 2014): this microphysics parametrisation has an explicit nucleation of water droplets (naCCN) and activation of ice particles (naIN) by aerosols.

Two different simulations are run, differing in the aerosol load only: at low elevations, the POLLUTED experiment has aerosol concentrations in the atmospheric boundary layer about one order of magnitude larger than the PRISTINE simulation (see Supplementary Material, SM, section S2, Fig. S12 and S13). The way this has been obtained is explained in the following. Initial vertical profiles of aerosols are provided at each location in the domain. By using the option (`use_aero_icbc=false`), the Thompson aerosol aware microphysics scheme computes a fake surface aerosol emission flux from these profiles (see Fig. S11 in SM). The concentrations of both water-friendly and ice-friendly aerosols are updated at every time step and at any grid box taking into account advection, diffusion, and tendencies induced by the aerosol-cloud interactions. A zero-gradient lateral boundary condition (no flux) is applied on the coarser domain, and the constant fake aerosol emission flux at the surface provides the aerosol source, which is uniform in space and very low in the PRISTINE simulation and varies in the POLLUTED

simulations by more than two orders of magnitude as function of surface elevation (see Fig. S11 in SM). The parameter values that define those configurations in the aerosol aware microphysics routine are provided in section S2 of the supplementary material. The simulations have been configured so that the aerosols don't interact with radiation (`aer_opt=0`), allowing us to focus on the indirect effects only.

Indirect effects of aerosols include their influence on the radiation budget and on hydrology through their impact on cloud microphysical processes. For this reason the main variables analyzed in this work are temperature at two meters from the surface, cloud fraction and precipitation.

In the following, a cloud event at a given position in time and space is defined when at least one of the vertical levels admits cloud fraction equal or larger than 0.5. The number of cloud events for each pixel is then simply the count of cloud events during a considered time period. We verified that the precise threshold of cloud fraction chosen to define cloud events doesn't significantly impact the results (see supplementary material, SM, section S3, Fig. S14). Note that in principle this metric equally weights shallow and deep clouds. However, convective clouds at times can cover a relatively small surface area and might thus not be detected by this method, which in the inner domain requires a cloud cover of at least 8 km$^2$ within each cell. For this reason, to analyze the occurrence of convective events we also used the daily maximum upward velocity in the low to mid troposphere (i.e. at pressures higher than 400 hPa).

The statistical significance of the results has been assessed using the one-tail Student's t-test at the 95% confidence level: for temperature and precipitation the Student's t-test has been performed over the hourly datasets, while for cloud events the test has been done over the annual time series of the mean seasonal number of cloud events.

## 3  Results

Aerosol concentrations are particularly large in the Planetary Boundary Layer (PBL), as the surface input of aerosols is efficiently redistributed by turbulent mixing within the whole layer, and lower aloft. The PBL height is relatively shallow during winter months (on the order of $500 \pm 100$ m at noon, Fig. S6a in SM), and thicker during summer months, when the lower static stability of the atmosphere favors rising thermals (mean summer PBL height at noon over the entire domain is $1300 \pm 500$ m, Fig. S6b in SM). Given the very different tropospheric dynamics in the different seasons, we present results for winter (DJF) and summer (JJA) separately. Considering that in our simulations aerosol input depends on elevation only, and not on surface type, high injection levels are provided over the low altitude continental areas as well as over the sea. In the POLLUTED simulation, this leads to a high aerosol concentration over the Ligurian and Adriatic seas, which is not particularly relevant for understanding the effects of urbanization and of anthropic activities. In the following, we thus present results for land points only. Figures including marine areas are shown and briefly discussed in the supplementary material.

The difference in the seasonal mean number of cloud events between POLLUTED and PRISTINE simulations is shown in Fig. 2c and d. Larger aerosol concentrations are associated with more clouds. This response is related to the well known fact that droplets nucleate over cloud condensation nuclei (CCN) and that a larger number of cloud droplets inhibits their growth to the size where they precipitate, leading to longer cloud lifetime (Albrecht, 1989; Christensen et al., 2020). However, the change in

the number of cloud events is both dependent on topography and on season. The cloud cover increase is particularly large at low elevations during DJF. During JJA, no significant differences in the statistics of cloud events are present in the Po Valley, and the increase of cloud cover is limited to high elevation areas and to some coastal regions.

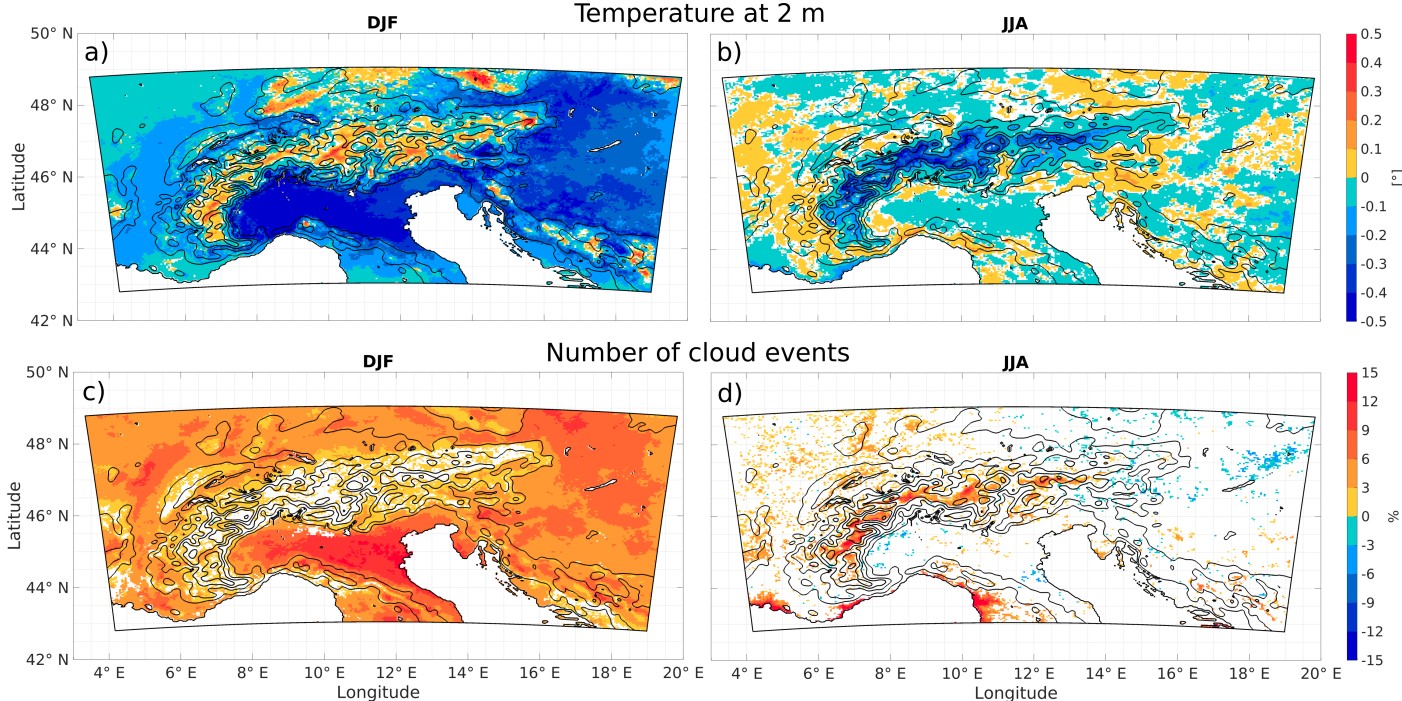

**Figure 2.** *Seasonal mean difference in 2 m temperature in Winter (a) and Summer (b). Seasonal relative variation of number of cloud events in DJF (c) and in JJA (d). Coloured pixels represent points that are significant at the 95% confidence level. Altitude isolines are shown every 500 m.*

We now turn our attention to the difference in seasonal mean temperature at two meters above the ground. Figure 2a and b indicates that temperature is generally lower in the POLLUTED run compared to the PRISTINE case, especially in winters.

However, a major feature emerges: in winters, over the mountains temperatures are actually higher in the POLLUTED run than in the PRISTINE one. Furthermore, lowlands are much colder in POLLUTED than in PRISTINE during winters, while during summers the differences in the lowlands are minor and of contrasting signs. Overall, the POLLUTED-PRISTINE 2 m temperature difference increases with surface elevation during winters (from negative to positive values), and decreases with surface elevation during summers (becoming very negative at high elevations).

The spatial pattern of the temperature anomaly is consistent with the difference in cloud coverage: cloud scattering decreases shortwave radiation from the sun reaching the ground, resulting in an overall surface cooling. We indeed verified that the seasonal mean shortwave radiation at the ground is always smaller in POLLUTED compared to PRISTINE. However, it remains to be clarified why a non significant change in cloud cover during winters at high elevations is associated with a

temperature increase in POLLUTED, and to explain the origin of the spatial heterogeneity of the cloud cover response, which appears to be related to orography and to land-sea contrasts. For this reason, we analyze the diurnal cycle of the differences in cloud events and in near surface temperature at different altitudes. The differences of mean hourly temperature and of mean cloud events averaged over ranges of altitude for land points only are shown in Fig. 3.

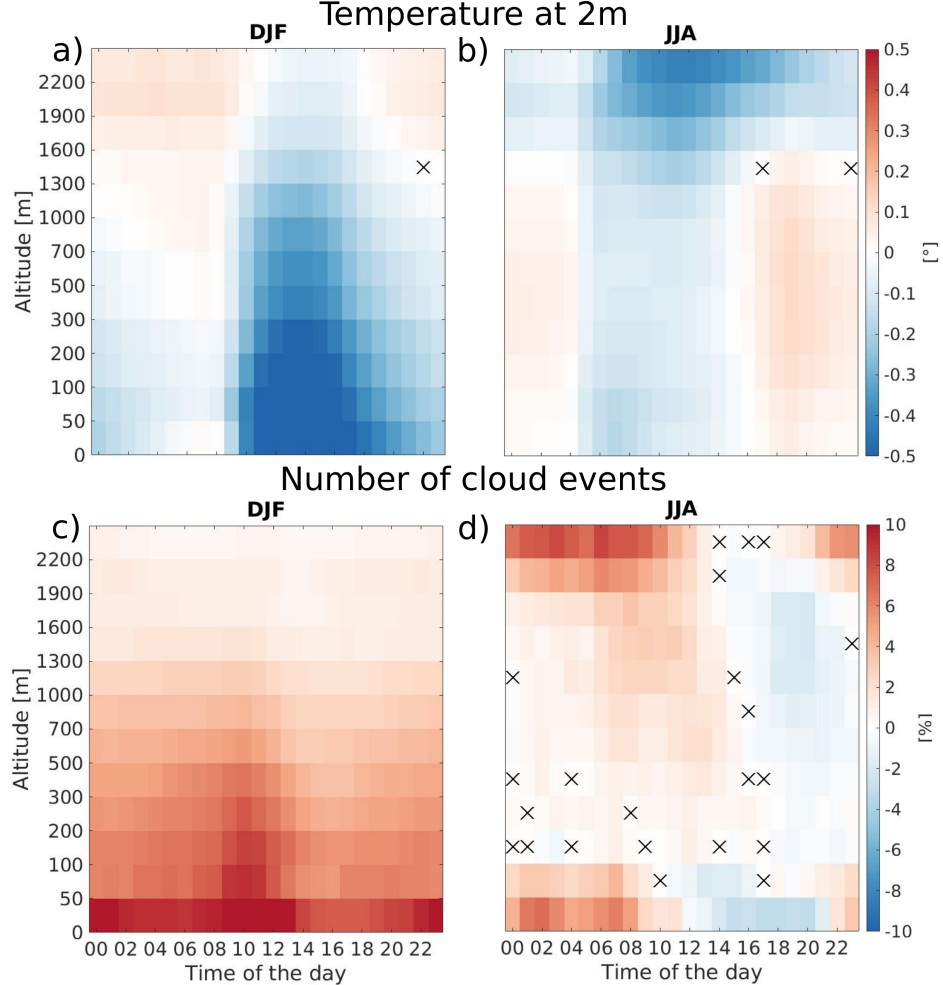

**Figure 3.** *Daily cycle at local time of the difference between POLLUTED and PRISTINE of the mean hourly temperature at 2 m in DJF (a) and in JJA (b). Daily cycle of the relative variation between POLLUTED and PRISTINE of the mean hourly number of cloud events in Winter (c) and in Summer (d). Crosses represent points that are not significant at the 95% confidence level.*

During winters, we note the presence of a relatively strong diurnal cycle in near surface temperature differences between POLLUTED and PRISTINE, which has different characteristics at different elevations (Fig. 3a). The relative increase in cloud cover has a strong elevation gradient, with cloud event number being 10% larger in POLLUTED than in PRISTINE over land close to sea-level, and only 1% larger over mountain areas above 2000m (Fig. 3c). This first result is related to the

fact that aerosol emissions strongly depend on elevation in the POLLUTED experiment, and that the stratification of the lower atmosphere prevents a significant transport of pollutants at high elevations, that remain relatively aerosol-free, with aerosol concentrations just slightly larger than in the PRISTINE case. The larger amount of clouds reduces shortwave radiation reaching the ground and near surface temperature during the daylight hours (see Fig. 3a), and it also reduces outgoing longwave radiation, allowing for the warming observed at high elevation during night hours (see section S4 of the supplementary material, Fig. S17). At low elevations the effect on shortwave radiation dominates, resulting in a strong reduction of daily averaged near surface temperature (-0.3 °C at elevations between the surface and 100m), while at high elevations the effect on longwave radiation dominates, resulting in a weak increase of daily averaged near surface temperature (+0.03 °C at altitudes above 1900m, see Fig. 2a). We'll get back to this difference in the discussion section.

During summers the daily cycle of the temperature variation (Fig. 3b) has a different behaviour than in winter: in lowlands there is a weak negative anomaly in the first part of the day, and a weak positive anomaly starting from late afternoon and through the night, resulting in a daily mean temperature difference between POLLUTED and PRISTINE runs close to zero (see Fig. 2b), while at high elevations the anomaly is negative throughout the day, with largest values during daytime. Cloud events, rather than being increased throughout the day as in winter, are typically increased during the night and the first hours of the day, and are generally suppressed in the afternoons (see Fig. 3d), resulting in a relatively small daily mean response. This suggests that there is a reduction in the number of convective events, which are particularly frequent in late afternoons. To explore this aspect, we show in Fig. 4 the relative difference in the number of strong updrafts (vertical velocity larger than 3.5 m/s (LeMone and Zipser, 1980; Kahraman et al., 2017)) between POLLUTED and PRISTINE as function of elevation. The figure shows a wide range of responses, consistent with the fact that convection is a highly intermittent process and that the interannual variability of convective storms at a specific location is large. Still, it can be seen that while the averaged updraft occurrence in the two simulations is similar at low elevations, at higher altitudes POLLUTED has less convective events than PRISTINE. The updraft velocities are also reduced in presence of many aerosols (Fig. S16 in SM).

Finally, we show in Fig. 5 the precipitation response, as function of surface elevation, which indicates a general drying. During winters, aerosol load weakly affects precipitation at sea level, then the relative variation of rainfall increases with elevation until about 1000m above sea level (asl) where seasonal mean precipitation is reduced by about 7% in the POLLUTED run compared to PRISTINE, and at higher elevations the difference between the two runs decreases again. During summers, the relative variation of precipitation is nearly monotonic with surface elevation, and the POLLUTED run is on average about 20% drier than the PRISTINE simulation above 2000m asl.

The results presented in this section clearly indicate that the climatic response to aerosols, through their indirect effect, is complex, dependent on topography, and different in different seasons.

## 4    Discussion and Conclusions

In the previous section it has been shown that the indirect effect of aerosols can lead to either warming or cooling at the surface. The response depends on surface elevation and season. Here we discuss the physical processes that are responsible for

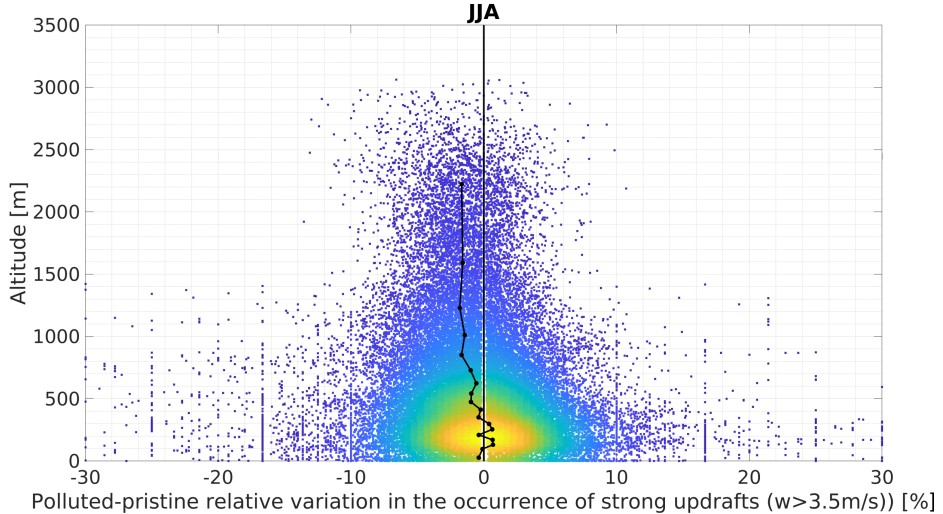

**Figure 4.** *Relative variation of the number of events with at least 3.5 m/s upward vertical velocity in function of elevation in JJA. The black line indicates the mean relative variation in 18 classes based on grid cell surface elevation, defined to have the same number of data in each class. Colors represent the density of the points in the GAR from yellow (high density) to blue (low density).*

this complex response, which has been summarized in Fig. 6.

During winters, aerosols are concentrated in lowlands, as the shallow PBL traps them near the surface. They increase the number of cloud droplets, resulting in longer lived clouds which affect both net shortwave and longwave radiation at the surface. Net shortwave radiation depends on surface albedo, being smaller over the mountains where snow cover reflects a large fraction of solar radiation, and larger over the darker flatlands. The effect of the increased cloud cover during winter
is thus dominated by the reduced net incoming shortwave radiation at the surface in lowlands, while it is dominated by the increased downward longwave radiation over the mountains. Near surface temperature is reduced in presence of higher aerosol concentration during daytime in both cases, but with much larger anomalies at low elevations than at high elevations, where indeed the aerosol load is smaller than in the valley and the albedo is larger. The reduced daytime soil temperature persists over night at low elevations, preventing the air temperature anomaly from becoming positive during night. At high altitude, however,
the daytime reduction is very small and does not last over the night, when the weakly increased cloud cover limits the infrared energy loss and leads to warmer temperatures. The longer lifetime of clouds associated with a larger number of CCN delays the onset of precipitation. While this does not really affect winter rainfall over flatlands, it leads to reduced precipitation where orographic clouds form (see Fig. 5), i.e. at medium elevations, especially on the upwind slopes (see Fig. S15 in supplementary material).

The summer dynamics is very different from what has been described so far. Surface warming by insolation leads to a thick PBL and, consequently, to relatively high concentrations of aerosols even at high elevations (see Fig. S6b, S12b and S13b in supplementary material). It also favors the development of convective storms, that are characterized by thick clouds that rapidly precipitate.

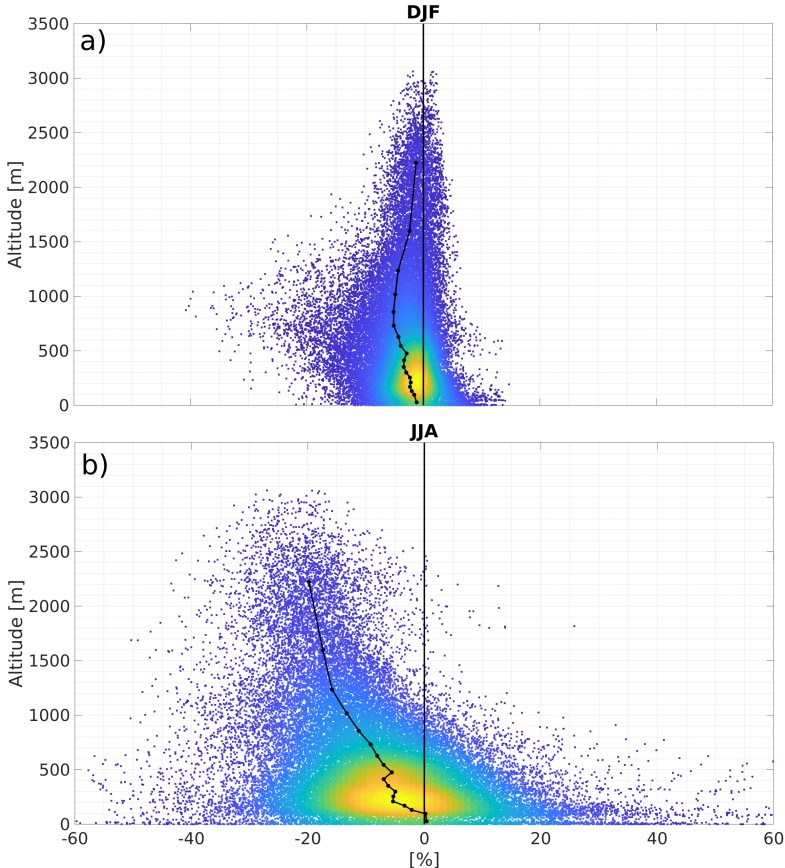

**Figure 5.** *Relative variation of the mean seasonal precipitation in function of elevation in DJF (a) and in JJA (b). The black line indicates the mean relative variation in 18 classes based on grid cell surface elevation, defined to have the same number of data in each class. Colors represent the density of the points in the GAR from yellow (high density) to blue (low density).*

Convective cloud evolution is very fast, characterized by large supersaturation values that lead in a short time to big raindrops. Previous research indicates that aerosols can modify convective cloud evolution through effects on cloud microphysics and dynamics that involve complex processes dependent on their chemical composition and on the environmental conditions, leading to contrasting results (Khain et al., 2005; Nishant et al., 2019; Zhang et al., 2020; Abbott and Cronin, 2021; Jiang et al., 2018; Khain et al., 2008; Fan et al., 2016). The short lifetime of convective clouds and their limited occurrence in the mid-latitude region under study imply that the variations in convective cloud cover in response to aerosol loading are small and do not significantly modify the daily mean insolation. For those reasons, we call convective clouds Aerosol-Independent Clouds (AIC) to imply that their daily mean radiative effects are not substantially affected by pollution, and to distinguish them from the Aerosol-Dependent Clouds (ADC) linked to synoptic scale disturbances and low level clouds.

During summer, both types of clouds can be present, with the Aerosol-Independent Clouds being concentrated in late afternoons. A tiny increase in ADC occurs and dominates the cloud cover variation at night and in the first hours of the day in the

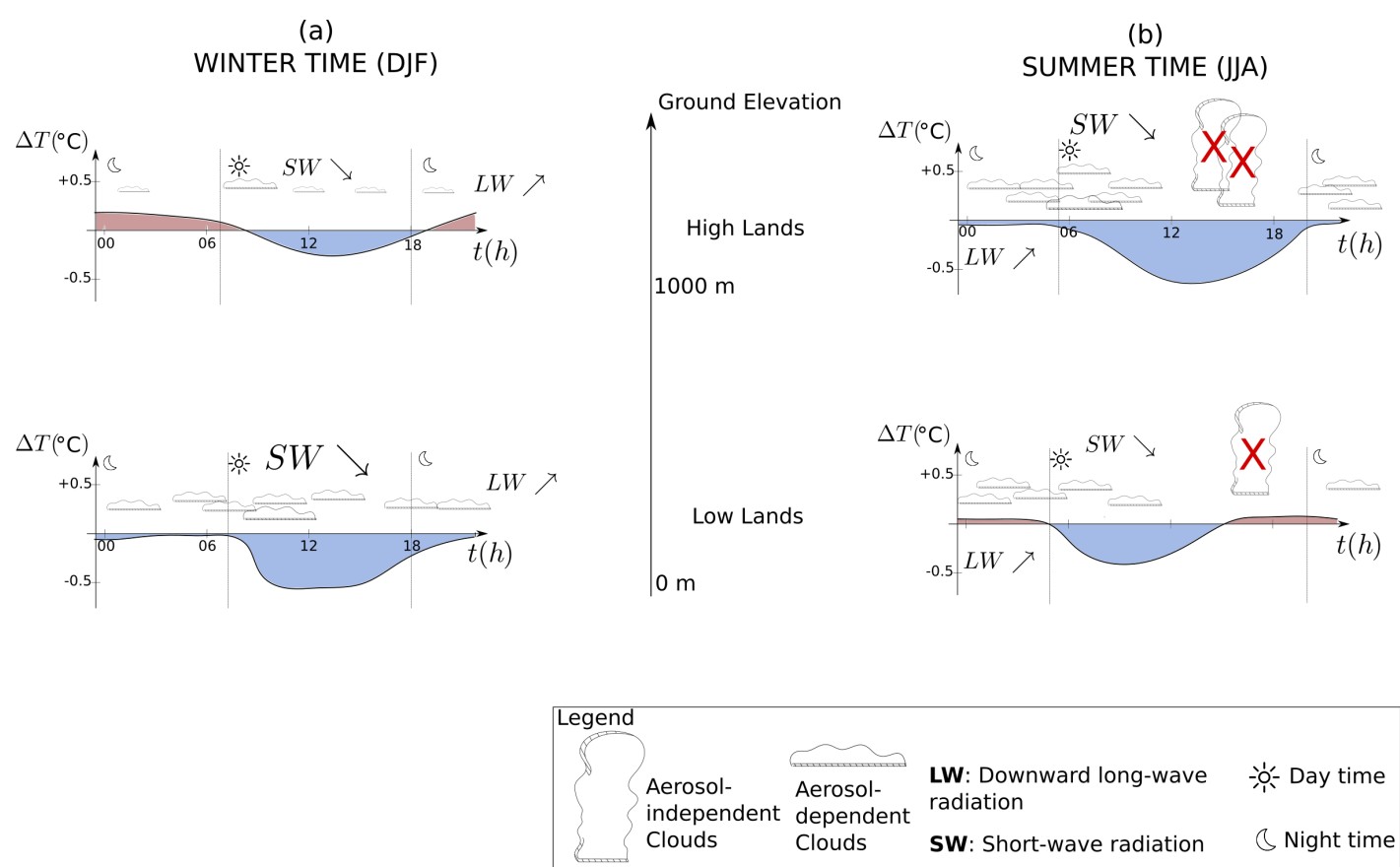

**Figure 6.** *Concluding sketch representing the daily cycle of the anomaly features in a POLLUTED environment in lowlands and highlands for (a) winter time (DJF) and (b) summer time (JJA). The arrows represent the change with respect to PRISTINE environment: upward arrows indicate an increase, while downward arrows a decrease. Colors blue and red represent negative and positive anomaly of temperature, respectively. Red crosses represent the suppression of the objects to which they refer.*

POLLUTED run compared to the PRISTINE one. This generates a small near surface temperature reduction that lasts until midday, slightly inhibiting convection and limiting its occurrence in the afternoon. The effect is to reduce the development of AIC (and of precipitation) and thus increase solar radiation at the ground: the negative temperature anomaly weakens in the high altitudes and it becomes positive in lowlands from late afternoon through the night.

At high altitudes clouds are more frequent, as they form when the winds are diverted upwards along the slopes of the mountains. Those orographic clouds are ADC. They are more frequent in POLLUTED and induce a strong cooling of the near surface air by filtering shortwave radiation. The negative anomaly in temperature is strong and persists over the night. Even at high elevation this cooling reduces the occurrence of convection, and limits the number of convective cloud events. Averaged over the whole day, however, there is an increase of cloud cover as the increase in ADC dominates over the decrease of AIC. This mechanism of suppression of convective clouds by aerosols is due to the decrease of temperature at the ground and

consequently to the increase of the air column stability. It has already been studied over Southern China (Zhang et al., 2020) and it is in agreement with the findings of Da Silva et al. 2018. It should be noted that opposite effects of aerosols on convective clouds have been described in studies referring to the tropics (e.g. Nishant et al., 2019; Abbott and Cronin, 2021), where indeed the convective instability is not sensitive to small variations in surface temperature.

The interpretation of the results in this study has been done without any attempt to separate the effect of aerosols as cloud
condensation nuclei and as ice nuclei. It is known that the effects of CCN and IN on clouds and precipitations can be very different: while it is generally thought that in warm clouds the increase in particulate increases the cloud lifetime and delays the onset of precipitation (e.g. Albrecht, 1989; Christensen et al., 2020), studies on mixed phase clouds indicate that the increase in ice nuclei could result in earlier and stronger precipitation (Zeng et al., 2009b, a; Deng et al., 2018; Yang et al., 2020). Further studies varying the concentration of CCN and IN separately in this region of complex orography will shed light on their relative
role in cloud formation, duration, and precipitation. The results could also depend on the size of CCN, a topic that has not been investigated in this study: Van Den Heever and Cotton 2007 showed that while larger loads of CCN can reduce precipitation, giant CCN can actually increase it. Furthermore, sensitivity studies to the magnitude of the surface aerosol fluxes and their dependence on surface elevation could be valuable, considering the complex and heavily nonlinear processes at play.

It should also be noted that the set up of our numerical experiments does not allow to properly account for small scale processes,
such as turbulent mixing. For instance, the evaporation-entrainment feedback is a process that has been described to occur on the edge of non-precipitating cumulus clouds and that can favor the evaporation of droplets at large aerosol concentrations (Jiang et al., 2006; Small et al., 2009). The setup we used in our experiments cannot account for those small scale processes.

The design of this work overlooked aerosol direct and semi-direct effects on radiation, that could modify, even substantially, the climatic response. However, modeling studies of this kind allow us to separately consider each effect and understand it.
Further work will be needed to analyze other aerosol effects on climate and to determine whether the total effect, through all the different processes, is merely a linear sum of each of them or not.

*Data availability.* The datasets generated and analysed during the current study are available from the authors on reasonable request.

*Author contributions.* A.N., A.P. and C.P designed the experiments; A.N. performed the simulations; A.N. performed the analysis and prepared the figures with the help of F.D.; C.P. conceived the project; A.N., F.D. and C.P. wrote the paper; all authors interpreted the results.

*Competing interests.* The authors declare that they have no conflict of interest.

*Acknowledgements.* The authors thank the two anonymous reviewers for their meaningful comments. The authors would like to thank Gregory Thompson at NCAR for helping to use the Thompson aerosol-aware scheme. Thanks are due to LRZ Supercomputing Centre, Garching, Germany, SuperMUC Petascale System, project-ID: pr62ve and to the HPC resources of CINECA under the allocations mBI20_AmbCo, mBI21_AmbCo and mBI22_AmbCo attributed through the CINECA-UniMiB convention. Both HPC resources allowed the execution of the runs and the subsequent analysis. This work is an outcome of Project MIUR – Dipartimenti di Eccellenza 2018–2022. The authors acknowledge support from the project JPI Climate Oceans EUREC4A-OA. FD has been partially supported by HPC-TRES grant number 2020-10 and by Piano Operativo Nazionale "Ricerca e Innovazione", Italian Ministry of University and Research, RTDAPON-150.

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
