# Peer review of "Aerosol indirect effects in complex orography areas: a numerical study over the Great Alpine Region"

_Atmospheric Chemistry and Physics, 2021_

## Referee Comment (RC1)

General comments

This work aims to identify the impact of aerosol-cloud interactions (indirect effect) over the Alpine region on temperature, precipitation and cloud fraction and also explore how this impact varies with elevation. It uses the WRF model to conduct two sensitivity experiments with low and high aerosol concentrations. The scientific question is very interesting. The paper is concise and well structured. Conclusions are substantial and the interpretation of physical properties is well discussed and summarized in a graph.

In general I believe that the methodology is sufficient for the scope of the paper. However I would like to see some more information in the methodology regarding some technical aspects of the experiments and discuss some issues. Also, I think some additional plots regarding some of the physical processes-impacts would be of value (even in the supplement). To summarize, it is a pleasant paper that I would recommend for publishing after some minor additions.

Specific comments

1. The WRF model is driven in the experiments by a GCM (EC-EARTH). Since this is a sensitivity study for a past period, I would expect that the standard would be to use reanalysis data to conduct the experiments. This would safeguard in the case the GCM has some major biases in the region and time chosen and fails to capture the climatology. I would be interested to know whether there is a specific reason for choosing the GCM instead of reanalysis. I understand that we are mainly interested in the differences between the polluted and the pristine experiments and not whether they sufficiently capture the climatology, so this is not a huge deal. However, since we are interested in the Alpine region it would be nice to know whether the experiments do capture the basic features of the climate there. Maybe a quick qualitative validation with some plots of observational/reanalysis in the supplement next to the control run?

2. Is there a specific reason for choosing the specific time period of 1979-1983 for the experiments?

3. I do not have any major issues with the domain setup. However, I would prefer the outer (larger) domain to be bigger. I would extent it more to the west, so that the distance between the outer and inner domains would be larger. In the current setup, any disturbance coming from the west would enter, pass through the relaxation zone and only after a few grid points would reach the inner domain. I don't know whether it has the distance to be sufficiently analyzed. Also, I would probably extend it more to the south to fully capture the Ligurian Sea thus better capture any possible cyclogenesis events. Just a few thoughts for future consideration.

4. I would like to see some more technical information in the methods regarding the WRF setup, like the number of vertical levels and the top pressure level used. Especially, in a highly mountainous regions this could be important.

5. As far as I understand you used the 'use_aero_icbc=false' option in the namelist regarding the Thompson aerosol aware mp. It would be interesting for WRF users to state this in the methodology. It can be in a parenthesis.

6. In the supplement S3 it is stated that some changes have been made in the code (orange highlights) to "better represent the aerosol concentrations in the Great Alpine Region." I would like a further elaboration on that. What is the rationality behind the changes?

7. Also in supplement S3, the aerosol number concentrations used in the experiments of the various variables (naCCN,naIN) are given. I think it would be important to state: Did you also change the numbers for the PRISTINE experiment from the original code? How did you choose the numbers for the POLLUTED run? Any possible references this was based on?

8. One major concern I would have regarding the Thompson aerosol aware mp in the 'use_aero_icbc=false' mode would be the aerosol concentrations over time. As far as I understand there are no aerosol coming from the lateral boundaries only the (fake) ground emissions. I don't know how this would affect aerosol concentrations throughout long term simulations. Have you checked how the aerosol concentrations over the domain for a specific experiment and season change throughout the years? Do they remain stable? Any large differences? If so, this could affect results from year to year. I would be really interested to know.

9. I think it would be interesting to note whether the aerosol concentrations produced in each experiment are plausible. For example, is the POLLUTED experiment something that can actually happen or is it an idealized case of unrealistically high aerosol pollution? Are the aerosol concentrations seen in the PRISTINE experiment a typical example of low pollution over the area? It would help to frame the overall research as being either mainly idealized or having ties with reality.

10. Page 4, lines 83-84. How much did you play with the cloud fraction threshold? If you could back up the cloud fraction threshold (in)sensitivity with a plot in the supplement the better (not necessary though).

11. In the Results section, impacts on shortwave and longwave radiation are mentioned. Since these are key to the interpretation of the impacts on temperature it would be nice to include figures on shortwave and longwave impacts in the supplement.

12. Figures 4 and 5 in the captions. You use the term "station altitude". If I am not mistaken, isn't this supposed to be "grid cell altitude"? Also please comment on what the colors mean in these figures.

13. Page 10.line 177. 'Convective cloud evolution…". Makes sense but I would like to see a reference.

Technical corrections

1. Abstract, line 4. Probably "the highest mountains" should change to "some of the highest mountains". Minuscule issue, however technically the highest mountain peak in Europe (the continent) is in Caucasus.

2. Introduction, page 1, line 18. "precipitations"→"precipitation"

3. Methods, page 2 line 52-53: The term "grid step" is understandable but I would probably change it to "grid resolution".

4. Page 7, line 144: "this aspects" → "this aspect"

5. Discussion. Page 9. line 167: "are reduced"→ "is reduced"

6. Supplement page 6: In the code segment there is the "niCCN3" variable whereas in the text below it is discussed as "naCCN3". I suspect one of the two must change.

---

## Author Comment (AC1)

Response to the reviewer n.1 of the manuscript by Napoli et al. 'Aerosol indirect effects in complex orography areas: a numerical study over the Great Alpine Region'

**Reviewer n. 1**

1. The WRF model is driven in the experiments by a GCM (EC-EARTH). Since this is a sensitivity study for a past period, I would expect that the standard would be to use reanalysis data to conduct the experiments. This would safeguard in the case the GCM has some major biases in the region and time chosen and fails to capture the climatology. I would be interested to know whether there is a specific reason for choosing the GCM instead of reanalysis. I understand that we are mainly interested in the differences between the polluted and the pristine experiments and not whether they sufficiently capture the climatology, so this is not a huge deal. However, since we are interested in the Alpine region it would be nice to know whether the experiments do capture the basic features of the climate there. Maybe a quick qualitative validation with some plots of observational/reanalysis in the supplement next to the control run?

Thank you for your suggestion. The reason for using EC-Earth output as the boundary conditions is because part of these simulations were done for another project where historical and future scenario simulations were compared. Having verified the consistency of the results with the observations and with the climatology of the same model forced with the reanalysis, we are confident that, for the aim of this work, the use of EC-Earth is appropriate.

Below we show a brief validation of the two experiments to verify that they capture the basic features of the climate of the Great Alpine Region. To evaluate the simulations, we carried out comparisons in the seasonal cycles of temperature (Figure R1-1a) and precipitation (Figure R1-1b) with the E-OBS dataset version 23.1 at 0.1°x0.1° (this new version provides daily gridded land-only observational dataset over Europe). To represent the climatological mean over the GAR, we analyze 30-years of the dataset. POL-LUTED and PRISTINE temperatures are interpolated over the E-OBS grid resolution using the bilinear method, while precipitation datasets have been interpolated using the First-order Conservative Remapping Method [1]. As we can notice, the seasonal cycle of temperature (Figure R1-1a) is well captured, showing high mean temperatures during the summer season and lower temperatures in winter. Precipitation (Figure R1-1b) is characterised by greater variability, nevertheless we can notice the increase of the mean precipitation during the spring season with a stronger decrease between August and September. The same model, forced with reanalysis data, was used in the work of [2] and showed consistent results. Spring season and June are characterized by a slightly overestimation of monthly precipitation compared to observational dataset, characteristic already observed and probably due to the complex terrain of this area [2, 3, 4].

**2. Is there a specific reason for choosing the specific time period of 1979-1983 for the experiments?**

Thanks for pointing this out. We realized that mentioning the years is misleading and totally irrelevant, considering that the runs are forced with the EC-Earth boundary conditions that certainly have not much in common with the precise years. We indeed have modified the manuscript removing the reference to the specific years used (line 50 in the Method section of the new manuscript: "Two 5-years long simulations have been run with initial and boundary conditions provided by the Earth System Model EC-Earth in a historical scenario on a 25 km horizontal grid"). Clearly, the choice of time period in this study is not significant because the purpose is to investigate the role of aerosols and not the effects of global climate change nor other climate variability patterns. We only need a long enough period in order to have statistically significant results, and to have an adequate number of years with realistic circulation patters.

3. I do not have any major issues with the domain setup. However, I would prefer the outer (larger) domain to be bigger. I would extent it more to the west, so that the distance between the outer and inner domains would be larger. In the current setup, any disturbance coming from the west would enter, pass through the relaxation zone and only after a few grid points would reach the inner domain. I don't know whether it has the distance to be sufficiently analyzed. Also, I would probably extend it more to the south to fully capture the Ligurian Sea thus better capture any possible cyclogenesis events. Just a few thoughts for future consideration.

Thank you for your comment about the domain setup, we will certainly keep it in mind for future work. It would certainly have been wiser to do what you suggest, but it's now too much computational demanding to rerun everything. We believe however that the results of the present study are not significantly impacted by this setup.

4. I would like to see some more technical information in the methods regarding the WRF setup, like the number of vertical levels and the top pressure level used. Especially, in a highly mountainous regions this could be important.

We added in the manuscript, in the updated Methods section lines 58-59, some more technical information. The text reads now: "The vertical structure of both domains consist of 50 terrain-following levels with a top pressure level set at 50hPa. The vertical resolution is finer near the ground (order of metre), while it coarser aloft (order of hundreds of metres)"

5. As far as I understand you used the 'use\_aero\_icbc=false' option in the namelist regarding the Thompson aerosol aware mp. It would be interesting for WRF users to state this in the methodology. It can be in a parenthesis.

Thank you, we added it in line 69 of the new version of the manuscript: "By using the option (use\_aero\_icbc=false), the Thompson aerosol aware microphysics scheme computes a fake surface aerosol emission.".

6. In the supplement S3 it is stated that some changes have been made in the code (orange highlights) to "better represent the aerosol concentrations in the Great Alpine Region." I would like a further elaboration on that. What is the rationality behind the changes?

The initial vertical profile of aerosols in the out-of-the-box microphysics 28 scheme has been designed to fit the Continental U.S. in which the near-surface value is found to exist within an idealized layer of approximately 200 to 1000 meters depending on starting elevation. The formulation tries to account for a very thin idealized layer height of tens of meters in high terrain above 2500 meters but closer to 1000 meters thick for grid points at elevations lower than 1000m. Considering the situation in the Great Alpine Region, we slightly modified the values such that the near-surface value of aerosols is found in a shallow layer for elevations above 1500m (rather than 2500m) and in a thicker layer for elevations lower than 600m. In any case, the memory of the initial profile is lost within an initial spin up, and only the initial near surface value is used to defined the forcing fields at any time step (fixed flux). We have added few lines in the revised supplementary material, section S3.

7. Also in supplement S3, the aerosol number concentrations used in the experiments of the various variables (naCCN, naIN) are given. I think it would be important to state: Did you also change the numbers for the PRISTINE experiment from the original code? How did you choose the numbers for the POLLUTED run? Any possible references this was based on?

Thank you for the question, this is absolutely an information that it is needed: we followed the procedure adopted by Da Silva et al. 2018, who chose values that "ensure that aerosol indirect effects emerge from the "natural noise" between MIN and MAX simulations" [5]. To this aim, Da Silva et al. define for the PRISTINE experiment profiles that are nearly independent of height (see Fig. R1-2) and for the POLLUTED experiment profiles that have the same upper level aerosol concentrations, but about one thousand times larger values near the surface in the valley. We used exactly the same values. The choice has been based on the observation that 10,000 /cc is a realistic value, measured in urban areas in the region under study (by counting particles larger than 0.01  $\mu m$  [6]). This information has been added to the new Supplementary Material version, section S3.

8. One major concern I would have regarding the Thompson aerosol aware mp in the 'use\_aero\_icbc=false' mode would be the aerosol concentrations over time. As far as I understand there are no aerosol coming from the lateral boundaries only the (fake) ground emissions. I don't know how this would affect aerosol concentrations throughout long term simulations. Have you checked how the aerosol concentrations over the domain for a specific experiment and season change throughout the years? Do they remain stable? Any large differences? If so, this could affect results from year to year. I would be really interested to know.

Excellent point. Thanks. Although there are no lateral fluxes of aerosols at the edges of the outer domain, the nested domain over which we analyze the results exchanges aerosols at the boundaries with the outer domain. We verified that the aerosol concentration in the planetary boundary layer does not significantly change through time. We show this in Figures R1-3 and R1-4, added here for your reference. Figure R1-4 shows the daily time series of the mean concentration over the PBL at 3 a.m. averaged over the domain of the POLLUTED experiment, for the first 34 months of the simulation (we don't have the same data for the remaining of the simulation). It can be seen than no clear trend appears. To explore weather a different behavior appears in the last couple of years in the simulation, we show in Fig. R1-3 the same variable over the five years of simulation, where only 1 datum per month is collected (at 3a.m. of the first day of the month). No clear trend appears in this time series either. Thus, we are quite confident that the results that we show do not have any dependency on long term changes in aerosol concentration. Regarding the seasonality that can be seen in Fig. R1-4, please notice that we only have data for 3 a.m. in the morning, when the PBL is very shallow and similar in thickness between summer and winter. For this reason, the timeseries does not reproduce the expected seasonality, with larger values during winter when the air column is more stable and lower values during winter when the PBL becomes very thick.

9. I think it would be interesting to note whether the aerosol concentrations produced in each experiment are plausible. For example, is the POLLUTED experiment something that can actually happen or is it an idealized case of unrealistically high aerosol pollution? Are the aerosol concentrations seen in the PRISTINE experiment a typical example of low pollution over the area? It would help to frame the overall research as being either mainly idealized or having ties with reality.

Thank you for the comment. The aim of this study was not to represent the real concentration of the aerosols in the Great Alpine region, although we have tried to describe the vertical profile in such a way that it better represents the area of our interest, as we replied to question number 6. In any case, the surface values corresponding to the POLLUTED experiment are values that have been observed in urban areas in Europe [6]. In that paper, seasonal mean values up to 80,000 /cc are reported (fig. 9). The low values used in our study (three order of magnitude smaller) correspond to concentration typical of the upper troposphere. In this sense, the PRISTINE simulation is probably not realistic. We decided to use those extreme values in order to maximize the effects of the changes and to explore the role of the surface emission.

10. Page 4, lines 83-84. How much did you play with the cloud fraction threshold? If you could back up the cloud fraction threshold (in) sensitivity with a plot in the supplement the better (not necessary though).

We verified that the difference in the total number of cloud events doesn't change significantly in function of the cloud fraction (Figure R1-5). We worked on three different thresholds, that are all shown in Figure R1-5: 0.1, 0.5 and 0.9. The figure has been added to Supplementary materials, new Section S6.

11. In the Results section, impacts on shortwave and longwave radiation are mentioned. Since these are key to the interpretation of the impacts on temperature it would be nice to include figures on shortwave and longwave impacts in the supplement.

Thank you for your comment about this aspect. Figure R1-6 shows the daily cycles of the difference of the mean hourly short-wave radiation (a, b) and downward long-wave radiation (c, d) in the two different seasons. We have added it to the supplementary material (new section S7) together with the climatological conditions for PRISTINE run added in section S2 (Fig. R1-7 and Fig. R1-8). We also made reference to the daily cycles of the difference of the mean hourly shortwave radiation and downward longwave radiation in the new version of the manuscript, Result section. These figures support our interpretation of the results, and make our point stronger. Thanks for the suggestion!

12. Figures 4 and 5 in the captions. You use the term 'station altitude'. If I am not mistaken, isn't this supposed to be "grid cell altitude"? Also please comment on what the colors mean in these figures.

Thank you for your advice: we have changed in Figure 4 and 5 the term "station altitude" and we have added the description about the colors in the scatter plots.

13. Page 10.line 177. 'Convective cloud evolution...". Makes sense but I would like to see a reference.

We modified these sentences to better clarify and we added some references in the new version of the manuscript:

"Convective cloud evolution is very fast, characterized by large supersaturation values that lead in a short time to big raindrops. Aerosols can modify convective cloud evolution through effects on cloud microphysics and dynamics that involve complex processes dependent on their chemical composition and on the environmental conditions, leading to contrasting results [7, 8, 9, 10, 11, 12, 13]. The short lifetime of convective clouds and their limited occurrence in the mid-latitude region under study imply that the variations in convective cloud cover in response to aerosol loading are small and do not significantly modify the daily mean insolation.

For those reasons, we call them Aerosol Independent Clouds (AIC) to imply that their daily mean radiative effects are not substantially affected by pollution, and to distinguish them from the Aerosol Dependent Clouds (ADC) linked to synoptic scale disturbances and low level clouds."

Figure R1-1: Seasonal cycle of temperature at 2m from the surface (a) and monthly mean precipitation over land points averaged over the GAR (b). Error bars represent the standard deviation of the monthly mean for E-OBS data calculated over land points of the entire domain and the full dataset (i.e, 30 years). The analysis has been done over the same domain for both POLLUTED and PRISTINE experiments and E-OBS dataset.

---

## Author Comment (AC2)

Response to the reviewer n.2 of the manuscript by Napoli et al. 'Aerosol indirect effects in complex orography areas: a numerical study over the Great Alpine Region'

**Reviewer n. 2**

1. Line 30: "avere anche" perhaps needs translation?

Thank you, correction done.

2. Line 35-36: You should add older references for the CCN suppression of conversion from cloud droplets to raindrops as well as reduced precipitation. This is a well-established effect far predating Thompson and Eidhammer 2014.

Thank you, we have added to the revised version of the manuscript: "increased Cloud Condensation Nuclei (CCN) slow the conversion of cloud droplets into raindrops [1, 2, 3, 4].".

3. Line 50: Why 1979-1983? More recent reanalysis products are available for model initialization and nudging.

Thanks for pointing this out. We realized that mentioning the years is misleading and totally irrelevant, considering that the runs are forced with the EC-Earth boundary conditions that certainly have not much in common with the precise years. We indeed have modified the manuscript removing the reference to the specific years used (line 50 in the Method section of the new manuscript: "Two 5-years long simulations have been run with initial and boundary conditions provided by the Earth System Model EC-Earth in a historical scenario on a 25 km horizontal grid"). Clearly, the choice of time period in this study is not significant because the purpose is to investigate the role of aerosols and not the effects of global climate change nor other climate variability patterns. We only need a long enough period in order to have statistically significant results, and to have an adequate number of years with realistic circulation patters.

4. Line 55: Your horizontal grid spacing is 4km, which might barely qualify as convection permitting. Can you cite some studies that justify 4km as convection permitting?

Simulations at this grid spacing lay in the "grey zone", where some small scale processes are resolved and others are parametrized. As the reviewer points out, this "grey zone" is really in need of further exploration, and the scientific community is working on performing systematic studies with the ultimate goal of developing a unified boundary layer turbulence parametrization valid across scales [5]. We are not there, yet, and for this study we adopt the practice of turning off the convection parametrization scheme while retaining the boundary layer turbulence parametrization scheme. Roberts & Lean (2008) [6] found significant benefits from turning off convective parametrization for grid spacing below 5 km, despite the fact that convective clouds cannot be properly resolved at these resolutions. Other views are present in the literature, suggesting that partially resolved convective motions should be damped as they are unrealistic [7]. The purpose of this study is clearly of a different nature and does not aim at improving knowledge in this regard: we want a relatively high resolution mainly to reproduce part of the complex orography in the area. While it is known that going to smaller grid spacing brings significant advantages in representing orographic regions, predicting events with small temporal and spatial scales [8, 9], we consider the 4 km resolution to be an appropriate choice for the purpose of this study based on preceding studies that used the same configuration and obtained good results when compared to lower resolution simulations and observational data [10]. We have added a short discussion on this in the Method section of the new version of the manuscript: "In the inner domain (4 km), no convective parameterization scheme is active. This configuration has already been used in previous works that compared results with lower resolution models and observational datasets [10], indicating that it is an acceptable compromise between the computational burden and the need of resolving smaller scales to predict events with small temporal and spatial scales [8, 9]."

5. Line 75: It would be helpful to show vertical profiles of aerosol number concentration during winter and summer over lowland and at elevation. This would help in interpreting the differences between pristine and polluted.

Thank you for the suggestion. Figure R2-1 shows the mean vertical profile over the entire domain of QNWFA in POLLUTED and PRISTINE experiment for ranges of altitudes in DJF (a) and in JJA (b): at higher altitudes during the summer season higher concentration are present compared to the winter season; for example at 2000m of altitude there is about one order of magnitude of difference between winter and summer profiles. The seasonal difference is not present in the PRISTINE experiment because we define an almost constant profile with the height. Figure R2-1 has been added to the Supplementary Material, Section S3.

Your initial differences of CCN number from 10 /cc to 10,000 /cc is a very large difference. 10,000 /cc is perhaps unrealistic for CCN sized aerosols.

Following the work described in Da Silva et al. 2018, we chose extreme values to "ensure that aerosol indirect effects emerge from the "natural noise" between MIN and MAX simulations" [11]. We use exactly the same values that they use. The choice has been based on the observation that 10,000 /cc is a realistic value measured in urban areas by counting particles larger than 0.01  $\mu m$  [12], fig. 9 (seasonal mean values up to 80000 /cc are reported in that paper). The low value of 10 /cc is a very low concentration typical of the upper troposphere and of the same order of magnitude of what originally included in the aerosol aware microphysics scheme for the upper levels. We added this information to the Supplementary Material, section S3.

How does the concentration evolve over time once the CCN flux scheme takes affect? What size aerosols are prescribed and what is their chemistry? Can aerosols from lowland be transported upslope to impact orographic clouds?

We thank the reviewer for having pointed us to the fact that not enough info on aerosols had been given in our manuscript. The information can be found in Thompson and Eidhammer 2014 [4], but it's worth to recap some details here. To limit the computational burden, the authors of the scheme chose not to a priori determine the specific aerosol types and chemical composition, but rather to define a *water-friendly* aerosol (with hygroscopicity parameter 0.4, considered as cloud nucleating) and an *ice-friendly* aerosol (non hygroscopic, considered as ice nucleating). Nucleation of cloud droplets occurs at a rate that depends on the number of water friendly aerosols, vertical velocity, hygroscopicity, and aerosol mean radius (0.04  $\mu$ m), assuming a lognormal size distribution with constant geometric standard deviation (1.8).

As we explain in the Methods section (lines from 70 to 76 of the manuscript), in these experiments there are no aerosols coming from the lateral boundaries, only the fake ground emissions that provide the aerosol source. Aerosols are advected by the wind field and removed by the microphysics scheme, through scavenging and as CCN.

We verified that the aerosol concentration in the planetary boundary layer does not significantly change through time. We show this in Figures R2-2 and R2-3, added here for your reference. Figure R2-3 shows the daily time series of the mean concentration over the PBL at 3 a.m. averaged over the domain of the POLLUTED experiment, for the first 34 months of the simulation (we don't have the same data for the remaining of the simulation). It can be seen than no clear trend appears. To explore weather a different behavior appears in the last couple of years in the simulation, we show in Fig. R2-2 the same variable over the five years of simulation, where only 1 datum per month is collected (at 3a.m. of the first day of the month). No clear trend appears in this time series either. Thus, we are quite confident that the results that we show do not have any dependency on long term changes in aerosol concentration. Regarding the seasonality that can be seen in Fig. R2-3, please notice that we only have data for 3 a.m. in the morning, when the PBL is very shallow and similar in thickness between summer and winter. For this reason, the timeseries does not reproduce the expected seasonality, with larger values during winter when the air column is more stable and lower values during winter when the PBL becomes very thick.

6. Line 77: Are aerosols radiatively active at all? You mention that they do not interact with shortwave radiation; but what about longwave radiation?

The option aer\_opt=0 allows to switch off the radiative effect of both short-wave and long-wave radiation. We have modified the line in the main text to clearly state this: "The simulations have been configured so that the aerosols of the Thompson's Aerosol Aware microphysics scheme don't interact with radiation  $(aer_opt=0)$ , allowing us to focus on the indirect effects of aerosols only."

7. Line 81: The wording in this sentence seems to need some correction starting with, "We define a cloud event....".

We have changed that sentence, that is now in lines 84-86 of the new version of the manuscript: "In the following, a cloud event at a given position in time and space is defined when at least one of the vertical level admits cloud fraction equal or larger than 0.5. The number of cloud events for each pixel is then simply the count of cloud events during a considered time period.".

8. Figure 3: For this figure and others as well, plotting things relative to local time would be more helpful than UTC when examining diurnal cycles. While the domain is close to UTC, examining things in local time would be more intuitive.

Thank you for the comment: all the figures in the main text and in the Supplementary Material are now in local time.

9. Line 129: How might your diurnal cycles change at elevation if you did not vary the emissions so dramatically with elevation? Is the amount of change in emissions with elevation justified?

To properly answer to this question, sensitivity runs would be needed. However, those runs are computationally heavy and we believe that this is not the right set-up to perform sensitivity experiments. It would be interesting to run idealized simulations in which this kind of effects can be studied. We have added a line in the Discussion to introduce this aspect: "Furthermore, sensitivity studies to the magnitude of the surface aerosol fluxes and their dependence on surface elevation could be valuable, considering the complex and heavily nonlinear processes at play". However, we believe that the processes described in the paper at the basis of the diurnal cycle observed are solid and well understood.

To support this claim, we added shortwave and downward longwave radiation diurnal cycle to the Supplementary material, and added here for reference (Figure R2-4): "During winters, aerosols are concentrated in lowlands, as the shallow PBL traps them near the surface. They increase the number of cloud droplets, resulting in longer lived clouds which affect both net shortwave and longwave radiation at the surface. Net shortwave radiation depends on surface albedo, being smaller over the mountains where snow cover reflects a large fraction of solar radiation, and larger over the darker flatlands. The effect of the increased cloud cover during winter is thus dominated by the reduced net incoming shortwave radiation at the surface in lowlands, while it is dominated by the increased downward longwave radiation over the mountains." In Summer, "At high altitudes clouds are more frequent, as they form when the winds are diverted upwards along the slopes of the mountains. Those orographic clouds are ADC. They are more frequent in POL-LUTED and induce a strong cooling of the near surface air by filtering shortwave radiation. The negative anomaly in temperature is strong and persists over the night. Even at high elevation this cooling reduces the occurrence of convection, and limits the number of convective cloud events. Averaged over the whole day, however, there is an increase of cloud cover as the increase in ADC dominates over the decrease of AIC."

10. Line 148-149: Why are updraft velocities reduced with aerosol loading? Many recent studies in the literature point to invigoration of updrafts in high aerosol situations unless there is stabilization due to aerosol radiative effects at play (but you are not simulating aerosol radiative effects). Will you provide mean vertical profiles of cloud droplet concentration in the convective clouds? Droplet numbers and size impact autoconversion rates, condensational growth rates, and riming rates; and these can impact latent heating rates and updraft speeds.

The potential for aerosol particles to impact the properties of deep convective clouds (higher updraft speeds) has been widely debated [13, 14]. As the reviewer pointed out, many studies in literature point to an invigoration of updrafts in high aerosols environment [15, 16], although invigoration, or suppression, depends on many factors [17]. For instance, clouds with warm or cold cloud base might respond differently [14] and the details of the microphysics scheme might be key. In our study, we don't focus on the convective precipitation per se, we don't analyze the time to precipitation in the convective clouds, nor the amount of rainfall in those events. The focus is here on the climatological effects of aerosols. The dominant mechanism, to explain the obtained result, is the increase in atmospheric column stability due to the surface cooling associated with a larger (non-convecting) cloud cover. We have added a discussion on this in the revised manuscript, that also describes the different effects aerosols have on convection and on daily mean insolation between tropical and extratropical regions: "This mechanism of suppression of convective clouds by aerosols is due to the decrease of temperature at the ground and consequently to the increase of the air column stability. It has already been studied over Southern China [18] and it is in agreement with the findings of Da Silva et al., 2018 [11]. It should be noted that opposite effects of

aerosols on convective clouds have been described in studies referring to the tropics [19, 20], where indeed the convective instability is not sensitive to small variations in surface temperature." As for the vertical profiles of cloud droplet concentrations, unfortunately, we do not have them, as we did

11. Lines 177-179: Here you state that convective cloud evolution is independent of aerosols. While aerosols typically do not control the convection, they can modify it depending on the strength of convection and aerosol concentration. There are complex interactions and feedbacks related to aerosols, precipitation, cold pools, and convective cloud lifetime that do permit aerosols to have impacts on the convection. It would be helpful to place your convection analysis in the context of additional cited literature involving aerosol impacts on convection. You might start with Alex Khain's paper classifying aerosol impacts on precipitation from various cloud types. https://doi.org/10.1175/2007JAS2515.1

not save them in the simulation output.

Thank you for your suggestion. We indeed clarified that convective clouds are directly affected by aerosols but their daily mean radiative effect is not significantly dependent on aerosol load. Different results are expected (and are in line with the literature e.g. [20, 19]) for tropical regions, where the air column stability is not significantly impacted by small surface temperature variations. We added few lines in the new version of the manuscript:

"Convective cloud evolution is very fast, characterized by large supersaturation values that lead in a short time to big raindrops. Aerosols can modify convective cloud evolution through effects on cloud microphysics and dynamics that involve complex processes dependent on their chemical composition and on the environmental conditions, leading to contrasting results [21, 19, 18, 20, 22, 23, 24]. The short lifetime of convective clouds.."

12. Line 187-189: Here you mention aerosol suppression of convection due to stability, but the aerosols in this study cannot directly impose such an effect. Further, van den Heever and Cotton (2007) do not discuss aerosol impacts on atmospheric stability. Please clarify your statement.

We realized we have not been clear enough. We mean that aerosols increase the cloud cover and this reduces shortwave radiation reaching the ground decreasing surface temperature. The indirect effect on the air column is thus an increase of stability. We have added in the manuscript a sentence to clarify the aerosol suppression of convection due to stability (lines 197-198 of the new manuscript): "This mechanism of suppression of convective clouds by aerosols is due to the decrease of temperature at the ground and consequently to the increase of the air column stability". Van Den Heever and Cotton [2007] don't discuss explicitly about the aerosol impact on atmospheric stability, but they point out that, in addition to microphysics and precipitation, aerosol can influence convective storm dynamics. In any case, we removed the reference to Van Den Heever and Cotton [2007] as potentially misleading. Thank you for the suggestion.

Figure R2-1: Mean vertical profile of QNWFA concentration for ranges of altitudes in Winter (a) and Summer (b) of the POLLUTED and PRISTINE experiments.